# OpenReview forum: "Scalable RF Simulation in Generative 4D Worlds"
_ICLR.cc/2026/Conference — Submitted to ICLR 2026_

### Official Review · Reviewer_Rnyj · 2025-10-29

**Soundness:** 3
**Presentation:** 3
**Contribution:** 2
**Rating:** 4
**Confidence:** 4

**Summary:**

This paper presents WAVEVERSE, a scalable framework that generates realistic RF sensing data by combining LLM-driven 4D world generation (text-based scene and motion synthesis) with phase-coherent ray tracing for physically accurate signal simulation, enabling diverse and high-fidelity RF datasets synthesis.

**Strengths:**

+ The integration of LLM-driven environment and motion generation with a phase-coherent ray-tracing simulator is thoughtfully designed and demonstrates strong engineering effort, providing a scalable pipeline for synthesizing diverse RF datasets.

+ The work is conceptually innovative in its vision of bridging generative AI and physical simulation, laying a solid foundation for future research that combines semantic scene synthesis with RF propagation modeling.

**Weaknesses:**

- Although the framework is well engineered, its core innovations build on existing foundations such as LLM-based 3D environment generation, SMPL-driven motion synthesis, and conventional ray-tracing techniques. The work would be stronger with clearer methodological advances that are unique to RF simulation, for example through learned physical priors, adaptive ray selection, or a formal analysis of phase coherence that distinguishes it from prior generative or physics-based systems.

- The idea is interesting and potentially impactful, yet the method inherently faces a fundamental limitation: it remains unclear how closely the simulated RF data align with real-world RF measurements. Because the 4D scene generation process creates entirely synthetic environments and motions that cannot be exactly replicated in the physical world, there is no definitive way to quantify the realism gap between simulated and real signals. Although the downstream case studies show encouraging performance gains, these results do not fully establish physical correspondence, leaving uncertainty about how faithfully WAVEVERSE captures real propagation characteristics.


- The path-based motion conditioning, while scalable, lacks explicit temporal control, which may result in unrealistic motion timing or velocity patterns that undermine physical plausibility in dynamic scenes. Moreover, the phase-coherent ray tracing, though conceptually well-motivated, lacks formal error analysis to verify its phase preservation accuracy.

**Questions:**

Please see the points raised in the Weaknesses section.

---

> ### Author Response · Authors · 2025-11-25
> **Author Response (1/3)**
>
> We sincerely thank the reviewer for their valuable time, insightful suggestions, and kind recognition of the potential impact of our work. We address the concerns regarding contribution, realism, and validation below.
>
> ---
>
> ### **Response to W.1: Innovation in RF Simulation and Comparison with prior RF simulation systems**
>
> We thank the reviewer for this comment and the opportunity to clarify the specific contributions of WaveVerse. We want to first highlight that WaveVerse introduces a new generation framework for RF signal simulation that transcends the inherent constraints of both prior learning-based and simulation-based methods.
>
> Existing **learning-based RF generation methods** like RF-Genesis and RF-Diffusion rely heavily on large, heterogeneous, and hardware-specific datasets for training. However, such datasets are rarely available in practice, especially for newly emerging sensor platforms with varying characteristics such as operating frequency, array geometry, or gain patterns. When trained on small or narrow datasets, these models often exhibit poor generalization to unseen scenes or novel hardware, and their lack of explicit physical modeling leads to outputs that mimic training data but may lack physical plausibility, resulting in weak performance.
>
> **Simulation-based methods** are similarly limited. Without generative assets, these systems are typically restricted to a handful of handcrafted environments or motions, severely limiting diversity. Moreover, most methods neglect the surrounding environment and miss higher-order multipath between humans and nearby objects. Their frame-by-frame ray tracing also causes reflection points to drift over time, breaking the spatial and temporal coherence needed for realistic phase behavior.
>
> In contrast, WaveVerse directly addresses these limitations by pairing generative 4D worlds with explicit geometry and physically grounded propagation. This hybrid design provides large-scale scene and motion diversity while preserving accurate multipath and stable phase evolution, and it generalizes to diverse radar configurations without retraining.  By unifying generative diversity with physical realism, WaveVerse achieves capabilities that neither learning-based nor simulation-based approaches can provide.
>
> These capabilities are enabled by our technical contributions on both the generative and simulation sides. **On the generative side**, we introduce a state-aware transformer with path-conditioning that ensures the generated motion conforms to the geometry and semantics of the scene. The model jointly conditions on both the path and the text description and outperforms baselines. **On the simulation side**, we propose phase-coherent ray tracing, which extends standard ray tracing to maintain consistent multipath geometry interaction and phase evolution across space and time for moving objects like humans and arbitrary radar configurations.
>
> (continued)

---

> ### Author Response · Authors · 2025-11-25
> **Author Response (2/3)**
>
> (continued from previous response)
>
> To highlight these advantages, in our revised manuscript, we have included the comparison between WaveVerse with the prior work in Sec. 4.4. For **Human Activity Recognition**, we have compared WaveVerse with **RF Genesis** by progressively augmenting the real dataset with 4x, 9x, and 19x simulated samples generated by each method.
> The table below reports the resulting HAR accuracy.
>
> |Method|Real only| +4x sim|+9x sim|+19x sim|
> |---|---|---|---|---|
> |WaveVerse|31.6%|49.8%| 61.4%|71.6%|
> |RF Genesis|31.6%|46.6%|55.8%|54.6%|
>
> WaveVerse consistently provides greater gains at every augmentation level. While RF Genesis yields some improvement at low augmentation ratios, but its performance plateaus when more simulated data is added. In contrast, WaveVerse continues to scale effectively, demonstrating that its physically grounded and environment-aware simulation produces higher-fidelity signals.
>
> Additionally, we have also included a comparison in the **High-Resolution Imaging** task. We note that existing learning-based RF generation methods such as RF Genesis **do not** support the rotating cylindrical radar array used in this case study. Thus, we compare with the **standard ray tracing**, which uses the same simulator configuration as WaveVerse but removes phase-coherent modeling. We augment the 1,000 collected samples with 1×, 2×, and 4× synthetic samples generated either by WaveVerse or by Standard RT and summarize the results below.
>
> |Metric|Method|Real only|+ 1x sim|+ 2x sim|+ 4x sim|
> |-|-|-|-|-|-|
> |MAE (↓)|WaveVerse|20.10|19.29|19.12|18.08|
> ||Standard RT|20.10|21.45|21.89|22.28|
> |Q-90th (↓)|WaveVerse|48.46|45.19|44.35|41.58|
> ||Standard RT|48.46|49.98|50.24|53.29|
> |PSNR (↑)|WaveVerse|26.96|27.66|27.69|28.47|
> ||Standard RT|26.96|27.01|26.85|26.89|
>
> WaveVerse clearly outperforms Standard RT across all metrics. Moreover, performance improves steadily as more WaveVerse-generated data is added, whereas standard ray tracing leads to diminishing or even negative returns at higher augmentation levels, suggesting that phase-incoherent simulation produces unreliable signals that do not benefit learning.
>
> In summary, WaveVerse establishes a new paradigm for RF signal simulation by integrating generative 4D world construction with physically grounded wave propagation. This unified framework, together with our technical innovations in scene-aware motion generation and coherent ray tracing, enables a level of diversity, realism, and generalizability that was not possible with prior approaches. We believe this direction opens new opportunities for scalable RF data generation, model training, and future research in RF sensing and simulation.
>
> ---
>
> ### **Response to W.2: Alignment with real-world RF measurements**
>
> We thank the reviewer for emphasizing the importance of aligning simulated signals with real RF measurements. We agree that this “realism gap” is a central challenge for generative systems. Since exact paired ground truth is unattainable, we instead assess alignment at the distribution level on the high-resolution imaging task, following standard practice in generative tasks.
>
> Specifically, we compute both *Fréchet Inception Distance* (*FID*) and the *Jensen–Shannon divergence of the TR margin* (*JS Div.*) between simulated and real signals. *FID* quantifies the distance between the feature distributions of real and simulated data, while *JS Div.* measures the gap between the confidence of a model on real data vs. simulated data. For both metrics, a lower value indicates better alignment. We train a U-Net on the RF imaging task and compute *FID* on features extracted at the bottleneck, and compute *JS Div.* following [1].
>
> Our results show that WaveVerse achieves an *FID* of 2.879 and a *JS Div.* of 0.365. These numbers are on par with those reported for strong generative models [2, 3, 1], and removing the modeling of phase coherence degrades performance substantially (*FID* 5.495 and *JS Div.* 0.430). Moreover, models trained on WaveVerse data transfer well to real measurements across tasks (Sec. 4.4), which also shows the fidelity of the simulated signals.
>
> We sincerely appreciate the reviewer’s suggestion, and we have added this result and discussion into our revised manuscript (the fourth paragraph of the High-Resolution Imaging (Sec. 4.4)).

---

> ### Author Response · Authors · 2025-11-25
> **Author Response (3/3)**
>
> ### **Response to W.3: Physical plausibility of generated human motion and the analysis of the phase preservation**
>
> We appreciate the reviewer giving us the opportunity to clarify these points. We address the physical plausibility of motion and answer the questions about the phase preservation in the separate parts below.
>
> 1). **Motion Plausibility**. We address the concern about the motion plausibility from the following two perspectives:
>
> *First*, our motion generation approach does not directly regress joint positions from text prompts. Instead, it operates within the discrete latent space of a VQ-VAE trained on large-scale motion capture data, which inherently encodes strong kinematic priors. By generating token sequences that are later decoded within this learned latent manifold, we effectively constrain the output to realistic human motions, significantly reducing the risk of physically implausible poses or unnatural joint configurations.
>
> *Second*, our method achieves an *FID* of 0.238, indicating that the synthesized motions closely resemble real ones in terms of plausibility, smoothness, and natural motion dynamics. To further evaluate physical realism, we have included two additional metrics under *Physical Plausibility* (Sec. 4.1 of the revised manuscript), *Skating Ratio* (foot-sliding) and *Bone-Length Variance* (skeletal stability). Our generated motions have a skating ratio of 0.067, very close to the value of 0.057 for the real dataset. The bone length variance remains as low as 1.78 cm², indicating highly stable skeletal geometry. Together, these results affirm that WaveVerse not only generates semantically meaningful motion but also maintains high physical fidelity in line with real-world motions.
>
> 2). **Phase Coherence Validation**. To rigorously assess phase accuracy, we conduct three experiments in Sec. 4.3 that explicitly rely on correct phase information for Range-FFT and Doppler-FFT processing. These benchmarks are intentionally designed to test whether our phase-coherent ray tracer maintains accurate phase behavior across time and space. In all cases, WaveVerse not only surpasses standard ray‑tracing baselines but also exhibits physically faithful phase behavior in the qualitative results, affirming its phase preservation.  Below, we explain the results from the Breathing Sensing benchmark.
>
> - **Setup:** We animate a human SMPL mesh using real breathing traces and simulate the resulting radar signals.
>
> - **Physics:** For FMCW radar, the received signal’s phase shift is linearly proportional to displacement, following the relation $\phi = \frac{4\pi d}{\lambda}$. Thus, accurately modeling the phase enables the reconstruction of the displacement signal $d$, which reflects chest motion due to breathing.
>
> - **Result:** WaveVerse successfully reconstructs the breathing signal with high fidelity, closely matching the ground-truth displacement and significantly outperforming the baseline (Figure 7). This demonstrates that our method preserves the correct phase evolution needed for fine-grained motion sensing.
>
> We further conduct experiments with a panoramic imaging setup for RF imaging and Doppler estimation of moving objects, with visualizations provided in Figure 6 in the main text and Figure 19 and 20 in our Appendix. These results provide evidence that our method well preserves the phase information. In addition, the distribution-level evaluations with collected data discussed in our response to W.2 show that RF signals simulated by our phase-coherent ray tracing align closely with real-world measurements.
>
> [1] Gong, Chen, et al. "Data Can Speak for Itself: Quality-guided Utilization of Wireless Synthetic Data." Proceedings of the 23rd Annual International Conference on Mobile Systems, Applications and Services. 2025.
>
> [2] Tian, Keyu, et al. "Visual autoregressive modeling: Scalable image generation via next-scale prediction." Advances in neural information processing systems 37 (2024): 84839-84865.
>
> [3] You, Zebin, et al. "Effective and efficient masked image generation models." arXiv preprint arXiv:2503.07197 (2025).

---

### Official Review · Reviewer_A1nB · 2025-10-30

**Soundness:** 2
**Presentation:** 2
**Contribution:** 2
**Rating:** 4
**Confidence:** 3

**Summary:**

This paper proposes a framework for scalable RF simulation inside generative 4D (space–time) worlds, featuring a language-guided world generator and a state-aware causal transformer for human motion (“WaveVerse”). It is novel to couple generative scene/motion synthesis with RF propagation to accelerate scenario creation at scale.

**Strengths:**

1. If the world/motion generator is language-guided, it could make scenario coverage and data diversity dramatically easier.
2. Potential to unify CV-style 4D generative assets with RF rendering, bridging two active communities.

**Weaknesses:**

1. No verified evidence here of RF accuracy vs. ground truth (ray tracing/EM solvers/measurements).
2. Treatment of multipath, diffraction, penetration, materials, etc, unclear.
3. Generators may induce distribution shift; need calibration showing RF outputs remain physically plausible under varied prompts.

**Questions:**

1. Do you calibrate generative geometry/materials to match measured RF responses? Any domain-gap mitigation (e.g., distribution alignment, correction nets)?
2. Sensitivity to prompt wording, motion model, mesh resolution, material catalog size; which factors dominate RF error?
3. How do you model material EM properties, antenna patterns, phase noise/CFO, timing offsets, and human-body scattering?

---

> ### Author Response · Authors · 2025-11-25
> **Author Response (1/3)**
>
> We thank the reviewers for their valuable assessment, comments, and the opportunity for us to provide the comparison with other methods and clarify the details of WaveVerse.
>
> ---
>
> ### **Response to W.1: Comparison with the ground truth signals**
> We thank the reviewer for highlighting the importance of validating RF accuracy against real measurements. In our setting, however, obtaining paired real-world ground-truth signals for each generated 4D world is not feasible, because the scenes and motions produced by WaveVerse do not correspond to any specific physical environment or trajectory that could be measured in practice.
>
> Given this practical limitation, we instead evaluate signal fidelity using **distribution-level metrics**, which are standard for assessing realism in generative modeling.  Specifically, we have included new quantitative evaluations measuring the similarity between simulated and real signals in our manuscript (the fourth paragraph of the *High-Resolution Imaging* (Sec. 4.4)). We report both Fréchet Inception Distance (*FID*) and the *Jensen–Shannon divergence of the TR margin* (*JS Div.*). *FID* quantifies the distance between the feature distributions of real and simulated data, while *JS Div.* measures the gap between the confidence of a model on real data vs. synthetic data. For both metrics, a lower value indicates better alignment. We train a U-Net on the RF imaging task, and compute *FID* on features extracted at the bottleneck, and compute *JS Div.* following [1].
>
> Our results show that WaveVerse achieves an *FID* of 2.879 and a *JS Div.* of 0.365. These numbers are on par with those reported for strong generative models [2, 3, 1], and removing the modeling of phase coherence degrades performance substantially (*FID* 5.495 and *JS Div.* 0.430). Furthermore, as discussed throughout Sec. 4.4, models trained on WaveVerse data transfer well to real measurements across tasks, which also demonstrates the fidelity of the simulated signals.
>
> ---
>
>
> ### **Response to W.2: Treatment of multipath, diffraction, penetration, materials**
> We thank the reviewer for bringing this up. We clarify these aspects below, and the revised manuscript now reflects these details in Sec. 3.1 and Sec. 5.
>
> 1). **Multipath:** Multipath is explicitly modeled in WaveVerse. We follow a standard ray-tracing formulation that simulates multi-bounce interactions with scene surfaces, including walls, floors, ceilings, furniture, and other objects. This produces the rich multipath structure characteristic of indoor RF propagation.
>
> 2). **Diffraction and Penetration:** Our current simulator is built on ray tracing with reflection modeling, which dominates indoor RF propagation and supports most RF sensing tasks. Consistent with prior simulators [4,5], diffraction around edges and penetration/refraction through materials are not modeled. In environments where these effects become dominant (e.g., narrow corridors with many sharp metallic edges), this simplification may contribute to a larger sim-to-real gap. The revised manuscript (Sec. 5) now explicitly discusses this limitation and outlines how WaveVerse can be extended with UTD-based diffraction and Fresnel-based penetration/refraction models. We appreciate the reviewer’s suggestion and agree that incorporating these advanced propagation models is a promising direction for future work.
>
> 3). **Materials:** We clarify our material modeling pipeline here, and the revised manuscript now includes this description in Sec. 3.1.
> WaveVerse adopts the ITU-R P.2040-2 standard to model dielectric properties, which provides frequency-dependent parametric models for permittivity and conductivity along with validated parameter sets for 14 common indoor materials. These parameters define physically validated dielectric constants used directly in our simulator.
> To expand material coverage, we sample objects from our asset library and prompt the LLM to propose additional material categories that follow the same ITU parametric form. We retain only categories whose dielectric parameters fall within physically valid ranges, resulting in a vetted library of 24 materials.
>
> During scene generation, the LLM does not infer dielectric values; it simply selects the most appropriate category from this curated physics-based library. This two-stage approach, physics-based parametric modeling followed by LLM-based categorization, ensures all material properties remain physically realistic while leveraging semantic cues from the object description.

---

> ### Author Response · Authors · 2025-11-25
> **Author Response (2/3)**
>
> ### **Response to W.3: Distribution of the Generated 4D World and Its Impact on Simulated RF Signals**
> We appreciate the reviewer’s thoughtful observation about the possibility of distribution shifts introduced by the generative components and their impact on RF signal fidelity. We fully agree that checking alignment between simulated and real signals is essential for a generative RF framework.
>
> To assess this, we conducted additional distribution-level evaluations using *Fréchet Inception Distance* (*FID*) and the *Jensen–Shannon divergence of the TR margin* (*JS Div.*), as detailed in our response to W.1. These metrics are computed between simulated RF signals which are generated from a wide range of prompts and real collected signals. As previously discussed, these results indicate that the simulated RF signals closely match the distribution of real measurements, even when the underlying 4D worlds are generated from a wide variety of prompts.
>
> ---
>
> ### **Response to Q.1: Calibration for 4D world and simulated signals**
> We thank the reviewer for this thoughtful question. In our current experiments, we do *not* perform any geometry/material calibration, distribution alignment, or correction-net–based domain adaptation. Instead, we intentionally evaluate WaveVerse in a direct-use setting to test how well physically simulated RF signals transfer to real-world tasks without task-specific tuning.
>
> For the high-resolution imaging case study, WaveVerse automatically synthesizes a variety of indoor scenes from our 4D world generator. For the HAR case study, we generate motions that correspond to the target activities in the original training dataset. The RF signals produced from these generated 4D scenes are then used as is for training in our case studies.
>
> A key reason this works without additional calibration is that our simulation pipeline is tightly matched to the actual radar hardware. We follow the radar’s technical reference manual for antenna gain patterns and use the same carrier frequency, bandwidth/slope, sampling rate, and other configuration parameters as in the real system. Combined with our phase-coherent ray tracing, physics-grounded material modeling, and diverse scene generation, this alignment already yields realistic RF measurements that transfer effectively in both case studies.
>
> Nevertheless, we agree that incorporating lightweight refinement models is an interesting future direction (Sec. 5 of our revised manuscript).
>
> ---
>
> ### **Response to Q.2: Sensitivity of simulated RF signals to prompt, motion, mesh, material, and other components**
> We thank the reviewer for this question. In WaveVerse, the primary determinants of RF accuracy are the components inside the physics-based simulator, including phase-coherent ray tracing, the number of cast rays, recursion depth, antenna gain patterns, and the radar configuration (carrier frequency, slope, bandwidth, sampling rate). These factors directly control path diversity, multipath richness, attenuation behavior, and phase accumulation. For example, reducing the number of rays or limiting recursion depth immediately suppresses higher-order reflections, degrading the channel impulse response and imaging quality. Likewise, mismatching hardware configuration parameters would shift the predicted range/Doppler structure relative to real measurements. For this reason, we match all radar parameters tightly to the real hardware, as described in our response to Q.1, and observe stable and realistic signal behavior across experiments.
>
> In contrast, factors such as prompt phrasing, motion model variation, mesh resolution, or the size of the material library mainly affect which scenes or motions are generated, rather than the correctness of the RF simulation itself. These components influence the content and diversity of the 4D world, but do not materially alter the underlying propagation physics once the geometry, materials, and radar configuration are determined.

---

> ### Author Response · Authors · 2025-11-25
> **Author Response (3/3)**
>
> ### **Response to Q.3: Modeling of EM properties, antenna patterns, and others**
> We thank the reviewer for raising these important modeling components. We clarify each part of the WaveVerse simulation pipeline below.
>
> 1.) **Material Electromagnetic Properties:** As detailed in our response to W.2, WaveVerse models material EM properties using the ITU-R P.2040-2 standard. This recommendation provides frequency-dependent parametric models for permittivity and conductivity for common indoor materials. We use these validated dielectric parameters directly in our simulator.
>
> 2.) **Antenna Patterns:** WaveVerse supports flexible antenna gain pattern configurations (e.g., isotropic, directional) by weighting ray amplitudes at the transmitter and receiver as a function of departure and arrival angles. These patterns are user-configurable at the simulation setup. For all experiments, we use the antenna pattern provided in the radar’s technical reference manual, ensuring alignment with the real hardware used in the datasets (as discussed in our response to Q.1).
>
> 3.) **Phase Noise, CFO, & Timing Offsets:**  Our framework allows phase noise, carrier-frequency offset, and timing offsets to be injected as optional signal-level impairments.
> In the experiments reported in the paper, we adopt the idealized configuration without explicit modeling of these impairments. One practical reason is that modern mmWave radar hardware has significantly reduced phase noise, CFO drift, and timing jitter, such that these effects typically act as small perturbations rather than major sources of error. That said, we acknowledge that for certain hardware-sensitive sensing tasks (e.g., fine-grained Doppler micro-motion analysis), explicit modeling of phase noise and CFO can become important. Incorporating these impairments into the generative simulation framework is therefore an interesting future direction, and we discuss this in Sec. 5 of the revised manuscript.
>
> 4.) **Human-Body Scattering:** WaveVerse incorporates the human body directly into the ray-mesh intersection process. The SMPL human mesh is treated as part of the scene geometry, and reflections from the body follow the same material-dependent specular–diffuse reflection model as all other surfaces. This ensures physically consistent scattering behavior and allows multipath involving the human body to be modeled coherently over time as the person moves.
>
> [1] Gong, Chen, et al. "Data Can Speak for Itself: Quality-guided Utilization of Wireless Synthetic Data." Proceedings of the 23rd Annual International Conference on Mobile Systems, Applications and Services. 2025.
>
> [2] Tian, Keyu, et al. "Visual autoregressive modeling: Scalable image generation via next-scale prediction." Advances in neural information processing systems 37 (2024): 84839-84865.
>
> [3] You, Zebin, et al. "Effective and efficient masked image generation models." arXiv preprint arXiv:2503.07197 (2025).
>
> [4] Cai, Hong, et al. "Teaching rf to sense without rf training measurements." Proceedings of the ACM on Interactive, Mobile, Wearable and Ubiquitous Technologies 4.4 (2020): 1-22.
>
> [5] Ren, Zhenyu, et al. "CASTER: A computer-vision-assisted wireless channel simulator for gesture recognition." IEEE Open Journal of the Communications Society 5 (2024): 3185-3195.

---

> ### Comment · Reviewer_A1nB · 2025-11-26
>
> W1
>
> The added metrics improve evidence of statistical realism, but they do not replace the need for scenario-level physical validation, even on a small set of controlled environments. This remains a significant limitation.
>
> W2
>
> Multipath modeling is clarified and aligns with typical ray-tracing pipelines, and use of ITU-R P.2040-2 and a vetted dielectric library is appropriate and reassuring. The explicit acknowledgment of limitations is appreciated; however, the practical impact on accuracy should be quantified.

---

> ### Author Response · Authors · 2025-12-03
>
> We are glad that most of the concerns have been addressed, and we appreciate the acknowledgment that the added quantitative metrics strengthen the evidence of signal realism. We are also encouraged that our clarifications on multipath modeling were well received and that our explanation of material modeling was regarded as “appropriate and reassuring.”
>
> For the remaining questions regarding **comparison with real-world signals** and **the impact of diffraction and refraction**, we conducted two additional comparisons summarized below.
>
> 1). **Validation with real-world signals in controlled environments.**
>
> We collect paired camera-radar data where a person walks forward and backward in a room. The synchronized camera video is processed with WHAM[1] to reconstruct a temporally consistent human mesh sequence. We then rebuild the surrounding environment (walls, floors) and simulate RF signals using WaveVerse, and compare them against the real measurements. The simulated and real range–time spectrograms achieve **28.63 dB PSNR** and **93.65% similarity** in energy distribution. This demonstrates that WaveVerse faithfully reproduces both structural motion patterns and amplitude dynamics of the measured signals.
>
> 2). **Validation with EM solvers and Impacts of diffraction and refraction.**
>
> To further validate the accuracy of our simulation pipeline and quantify the effects of diffraction and refraction, we compare WaveVerse with electromagnetic simulations from Ansys HFSS (High Frequency Structure Simulator) [2], a high-fidelity full-wave EM solver based on Maxwell’s equations. We adopt 16 setups across different previously generated scenes and sensor poses. For every configuration, we run HFSS simulations with and without diffraction and refraction effects enabled.
> - **Without diffraction/refraction**: WaveVerse matches HFSS with **33.57 dB PSNR** and **2.12% RMSE**.
> - **With diffraction/refraction**: Results remain strong at **31.25 dB PSNR** and **2.76% RMSE**.
>
> These findings show (1) WaveVerse closely approximates a gold-standard EM solver, confirming the correctness of our pipeline. (2) The small difference between the two HFSS settings indicates that diffraction/refraction has a limited influence in our target indoor scenarios.
>
> Importantly, each HFSS simulation requires over an hour, even for simple scenes, while WaveVerse produces comparable results in under one second. This highlights both the accuracy and the practical scalability of our approach.
>
> We believe these additional experiments provide direct scenario-level validation and a quantitative study of physical effects, addressing the remaining concerns in W1 and W2. Full details are provided in Appendix A.3 (p. 27), with a pointer added in Section 4.3. We thank the reviewers for their insightful feedback, which helped us strengthen both the experimental design and the clarity of our paper.
>
> [1] Shin, Soyong, et al. "Wham: Reconstructing world-grounded humans with accurate 3d motion." Proceedings of the IEEE/CVF Conference on Computer Vision and Pattern Recognition. 2024.
>
> [2] Stolarski, Tadeusz, Yuji Nakasone, and Shigeka Yoshimoto. Engineering analysis with ANSYS software. Butterworth-Heinemann, 2018.

---

### Official Review · Reviewer_wPPx · 2025-10-30

**Soundness:** 3
**Presentation:** 3
**Contribution:** 3
**Rating:** 6
**Confidence:** 4

**Summary:**

This paper addresses the critical challenge of acquiring large-scale, high-quality datasets for RF sensing by introducing WAVEVERSE, a scalable, prompt-based framework for simulating realistic RF signals in dynamic 4D worlds. The system integrates two main innovations: a language-guided 4D world generator and a physics-based, phase-coherent signal simulator. Experiments validate this approach, showing the phase-coherent simulation yields high-fidelity signals for beamforming and respiration monitoring. Case studies on high-resolution RF imaging and human activity recognition demonstrate that WAVEVERSE-generated data not only enables RF imaging simulation for the first time but also consistently improves downstream task performance in both data-limited and data-adequate settings.

**Strengths:**

Since RF Genesis, there have been few impressive papers in the field of RF simulation for quite some time. Overall, I hold a positive view of this paper for the following reasons:

- Case Study. I believe the most important aspect of evaluating RF simulation is the gain in real experiments. The current case study provides strong evidence of this. Based on this, I give a score of 6, and if the authors can address the weaknesses I have raised, I would be happy to increase it to 8 or higher.

- Introducing of Large Models. For RF simulation, introducing large models is an easily conceivable idea, but how to effectively introduce large models to improve system performance is not easy. The paper's approach to introducing large models is quite innovative and indeed effective.

- Signal Simulator. The newly designed signal generator demonstrates a certain degree of innovation.

**Weaknesses:**

1. My primary concern focuses on the RF baseline methods: As mentioned earlier, I believe the case study is very important. However, the current case study only demonstrates that the proposed method is effective compared to having no simulation data. But is it effective compared to data generated by other simulation methods? I suggest adding comparative experiments with data generated by other simulation methods.

2. The paper should provide more evidence that the motions generated by the LLM are physically plausible in terms of dynamics; it only proves that the motions are semantically aligned. A motion that "looks like slipping" does not mean it is physically possible. Similarly, the dielectric properties assigned by the LLM based on semantics have not been physically validated. The "realism" foundation of this framework (LLM input) is semantics-based, while its simulator (ray tracing) is physics-based; the paper does not validate the "physical realism" gap between these two domains.

3. There are some potentially overclaimed aspects in the paper:

(1) How to define "data-adequate." The paper claims to improve performance in data-adequate scenarios. Theoretically, when data is sufficient, the improvement from simulation data should be minimal. If the improvement is significant, then the data is not sufficient.

(2) "Data generation for RF imaging for the first time" is also strange. Although data generation for RF imaging is difficult, it is not necessarily the first time."

**Questions:**

See Weaknesses

---

> ### Author Response · Authors · 2025-11-25
> **Author Response (1/3)**
>
> We sincerely thank the reviewer for the thoughtful and supportive feedback, as it allows us to rethink how to better demonstrate the effectiveness and robustness of the proposed method, and eventually improve our work further.
>
> ---
> ### **Response to W.1: Comparison with other RF simulation baselines**
> We have added two additional simulation baselines and compared them with WaveVerse in the two case studies (Sec. 4.4 in the revised manuscript). We summarize the key findings below.
>
> 1). **Human Activity Recognition**
>
> We compared WaveVerse with **RF Genesis** using the same evaluation setup, progressively augmenting the real dataset with 4x, 9x, and 19x simulated samples generated by each method. The table below reports the resulting HAR accuracy:
>
> |Method|Real only| +4x sim|+9x sim|+19x sim|
> |---|---|---|---|---|
> |WaveVerse|31.6%|49.8%| 61.4%|71.6%|
> |RF Genesis|31.6%|46.6%|55.8%|54.6%|
>
>
> WaveVerse consistently provides larger accuracy gains at every augmentation level. While RF Genesis yields some improvement at low augmentation ratios, its performance plateaus when more simulated data is added. In contrast, WaveVerse continues to scale effectively, demonstrating that its physically grounded and environment-aware simulation produces higher-fidelity signals.
>
> 2). **High-Resolution Imaging**
>
> We note that existing learning-based RF generation methods such as RF Genesis **do not** support the rotating cylindrical radar array used in this case study. RF Genesis is designed and trained for a single radar with a fixed pose, and its neural architecture does not allow signal generation along a continuously moving radar trajectory. As a result, it is not applicable to our panoramic, rotating setup.
>
> Instead, we include a **standard ray tracing** (Standard RT) baseline, which uses the same simulator configuration as WaveVerse but removes phase-coherent modeling. This produces a simplified simulation similar to conventional ray tracing implementations (e.g., MATLAB-based).
>
> Using the same setup as our original case study, we augment the 1,000 collected samples with 1×, 2×, and 4× synthetic samples simulated either by WaveVerse or by Standard RT. Results are summarized below.
>
> |Metric|Method|Real only|+ 1x sim|+ 2x sim|+ 4x sim|
> |-|-|-|-|-|-|
> |MAE (↓)|WaveVerse|20.10|19.29|19.12|18.08|
> ||Standard RT|20.10|21.45|21.89|22.28|
> |Q-90th (↓)|WaveVerse|48.46|45.19|44.35|41.58|
> ||Standard RT|48.46|49.98|50.24|53.29|
> |PSNR (↑)|WaveVerse|26.96|27.66|27.69|28.47|
> ||Standard RT|26.96|27.01|26.85|26.89|
>
>
> WaveVerse consistently outperforms Standard RT across all metrics and augmentation levels. Moreover, performance improves steadily as more WaveVerse-generated data is added, while adding more Standard RT data yields no improvement and even degrades accuracy at higher ratios, suggesting that phase-incoherent simulation produces unreliable signals that do not benefit learning.

---

> ### Author Response · Authors · 2025-11-25
> **Author Response (2/3)**
>
> ### **Response to W.2: Physical Plausibility of LLM-Generated Motions and Materials**
>
> We thank the reviewer for raising this important point about the physical plausibility of the generated 4D worlds. We address this concern from the following three perspectives: (1) motion dynamics, (2) material dielectric properties, and (3) signal-level realism.
>
> 1). **Physical Plausibility of Generated Human Motion**
>
> We agree that semantic alignment alone does not ensure physically valid motion dynamics. WaveVerse addresses the concern in two ways:
>
> *First*, our motion generator does not regress raw joint coordinates from text. Instead, it operates in the discrete token space of a VQ-VAE trained on large human motion capture datasets, which encodes strong kinematic priors. During decoding, generated sequences are constrained to this learned manifold, which substantially reduces the chance of physically impossible poses or joint configurations.
>
> *Second*, our method achieves an *FID* of 0.238 between generated and real motion features, showing that the generated motions closely match real distributions in plausibility, velocity, and naturalness. To assess physical consistency, we have further provided quantitative evaluations of physical motion realism in the revised manuscript (*Physical Plausibility* under Sec. 4.1) with two additional metrics: *Skating Ratio* (foot-sliding) and *Bone-Length Variance* (skeletal stability). Our generated motions have a skating ratio of 0.067, very close to the value of 0.057 for the real dataset. The bone length variance remains as low as 1.78 cm², indicating highly stable skeletal geometry. Together, these metrics demonstrate that WaveVerse motions are not only semantically aligned but also physically plausible and consistent with real human dynamics.
>
> 2). **Material Dielectric Properties**
>
> We thank the reviewer for raising this important point. We clarify the material modeling pipeline here, and the revised manuscript now includes this description explicitly in Sec. 3.1.
>
> WaveVerse models dielectric properties following the ITU-R P.2040-2 standard, which provides frequency-dependent parametric models for permittivity and conductivity along with validated parameter sets for 14 common indoor materials. These parameters define physically meaningful dielectric constants that we use directly in the simulator.
>
> To extend beyond these 14 materials, we sample objects from our asset library and ask the LLM to propose additional material categories and follow the same ITU parametric model. We retain only categories whose dielectric values fall within documented physical ranges, resulting in a vetted library of 24 materials.
>
> During scene generation, the LLM is never asked to generate dielectric constants from scratch. Instead, it simply assigns each object to the most appropriate category from this curated library. This two-stage approach, physics-based parametric modeling followed by LLM-based categorization, ensures that all dielectric values remain physically validated while still allowing semantic description to guide material selection.
>
> 3). **Signal-level Realism**
>
> Finally, we use the simulated RF signals as an indirect but informative way to assess the fidelity of the generated 4D worlds. We quantify the distribution difference between simulated and collected RF signals on the high-resolution imaging task, using Fréchet Inception Distance (*FID*) and the *Jensen–Shannon divergence of the TR margin* (*JS Div.*). *FID* quantifies the distance between the feature distributions of real and simulated data, while *JS Div.* measures the gap between the confidence of a model on real data vs. synthetic data. For both metrics, a lower value indicates higher realism. Specifically, we train a U-Net model for RF imaging, compute *FID* on features extracted at the bottleneck, and compute *JS Div.* following [1].
>
> Our experiments show that WaveVerse achieves an *FID* of 2.879 and a *JS Div.* of 0.365. These numbers are on par with those reported for strong generative models [2,3,1], and removing phase coherence modeling degrades performance substantially (*FID* 5.495 and *JS Div.* 0.430). Moreover, as discussed in W.1, models trained on WaveVerse-simulated data transfer well to real-world signals in both imaging and HAR tasks. Taken together, these signal-level evaluations suggest that the generated 4D worlds in WaveVerse achieve a high degree of physical fidelity, beyond semantic realism alone.

---

> ### Author Response · Authors · 2025-11-25
> **Author Response (3/3)**
>
> ### **Response to W.3: Clarification on Claims**
>
> We thank the reviewer for pointing out these potential overclaims and have made the corresponding revision across the manuscript. First, the term “data-adequate” has been replaced with “data-rich” to better reflect our intended meaning: settings where a substantial amount of real data is available, yet additional synthetic data can still provide measurable improvements because the real dataset may lack sufficient diversity in poses, environments, or multipath structures. Second, we have refined our statement about RF imaging. Our contribution is not the first instance of synthetic RF imaging, but the first generative framework that supports flexible and mobile RF imaging configurations (including rotating arrays and variable apertures) that prior fixed-sensor generative approaches such as RF Genesis cannot model.
>
> [1] Gong, Chen, et al. "Data Can Speak for Itself: Quality-guided Utilization of Wireless Synthetic Data." Proceedings of the 23rd Annual International Conference on Mobile Systems, Applications and Services. 2025.
>
> [2] Tian, Keyu, et al. "Visual autoregressive modeling: Scalable image generation via next-scale prediction." Advances in neural information processing systems 37 (2024): 84839-84865.
>
> [3] You, Zebin, et al. "Effective and efficient masked image generation models." arXiv preprint arXiv:2503.07197 (2025).

---

### Official Review · Reviewer_hWWj · 2025-11-01

**Soundness:** 3
**Presentation:** 3
**Contribution:** 2
**Rating:** 6
**Confidence:** 4

**Summary:**

This paper presents WAVEVERSE, a framework that generate realistic RF signals from pure text prompts.
The system combines LLM-driven 3D scene and human motion generation with a phase-coherent ray tracing simulator that preserves spatial and temporal phase consistency, ensuring both physical fidelity and environmental diversity.
Experiments show that generated data (when combined with or replacing limited real-world RF datasets) significantly improves model performance in both data-limited and data-adequate settings, demonstrating its value as a scalable source of high-fidelity synthetic RF data.

**Strengths:**

1. Novel problem statement and formulation: RF signal in time series is really challenging.
2. Comprehensive dataset usage like Lai et al., 2024 for imaging and Singh et al., 2019 for activity recognition.
3. (Despite the overall work looks more like an itegration of existing techs rather than tacking theoretical challenges) The usage of LLM to align user's language with RF generation context and the implementation makes sense for potetial product use cases. The phase-coherence design aligns with SOTA.
4. Experiment setting makes sense to use generated data to boost current sensing task performance.

**Weaknesses:**

1. Limited technical depth: The work primarily integrates existing components (LLM-based scene generation, motion modeling and ray tracing) rather than introducing new theoretical insights or core algorithmic innovations (like NeRF^2 (MobiCom'23)[1] and RF Genesis(SenSys'23))[2].

2. Lack of real-data calibration: The generation workflow is not calibrated or partially trained with real RF measurements, which could clearly have had the chance to help bridge the domain gap and validate simulation fidelity.

3. Unclear inference overhead. Although the algorithm looks not very heavy, it's still needed for a multi-modal model/workflow to demonstrate the computation overhead.

4. Lack of real-world case evaluation / case-specific analysis. Despite it may be too harsh to criticize the work for lacking real-world experiments, it is important to dive into concrete real-world examples (materials or layouts for which the workflow performs worse, any external factors like surrounding WiFi/cellular signal may interfere, etc) to identify the true potential of the work.

[1] Zhao, Xiaopeng, et al. "Nerf2: Neural radio-frequency radiance fields." Proceedings of the 29th Annual International Conference on Mobile Computing and Networking. 2023.

[2] Chen, Xingyu, and Xinyu Zhang. "Rf genesis: Zero-shot generalization of mmwave sensing through simulation-based data synthesis and generative diffusion models." Proceedings of the 21st ACM Conference on Embedded Networked Sensor Systems. 2023.

**Questions:**

1. generalizability: In which circumstances would WAVEVERSE's performance degrade? How you intepret these cases?

---

> ### Author Response · Authors · 2025-11-25
> **Author Response (1/3)**
>
> We sincerely thank the reviewer for the thoughtful and encouraging assessment of our work. We appreciate the constructive comments regarding technical depth, calibration, and generalizability, which help us clarify our contributions and improve the presentation of the paper. We address each of these points in detail below.
>
> ---
>
> ### **Response to W.1: Technical contributions**
>
> We would like to highlight that the key contribution of WaveVerse is a new paradigm for RF signal generation that overcomes the fundamental limitations of prior learning-based and simulation-based approaches.
>
> **Learning-based RF generation methods** (e.g., RF Genesis and RF-Diffusion) require large, diverse, and sensor-specific RF datasets for training. However, such datasets are rarely available, especially for the emerging sensing modalities and novel RF hardware configurations that vary in wavelength, bandwidth, gain pattern, antenna characteristics, and array layouts. When trained on small datasets, these models fail to generalize beyond the few environments and subjects seen during training and remain tied to a single sensor configuration. In addition, because they do not explicitly model scene geometry or wave propagation, they often produce signals that resemble training samples but are not physically plausible, resulting in noticeably weaker performance than WaveVerse (Table 4 in our revised manuscript).
>
> **Simulation-based methods** are similarly limited. Without generative assets, they work with a small number of scenes or motions and therefore cannot provide the diversity needed for large-scale RF data generation. Moreover, most simulators ignore the surrounding environment and miss higher-order multipath between humans and nearby objects. Their frame-by-frame ray tracing also causes reflection points to drift over time, breaking the spatial and temporal coherence needed for realistic phase behavior.
>
> In contrast, WaveVerse directly addresses these limitations by pairing generative 4D worlds with explicit geometry and physically grounded propagation. This hybrid design provides large-scale scene and motion diversity while preserving accurate multipath and stable phase evolution, and it generalizes to *any* radar configuration without retraining. By unifying generative diversity with physical realism, WaveVerse achieves capabilities that neither learning-based nor simulation-based approaches can provide.
>
> These capabilities are enabled by our technical contributions on both the generative and simulation sides. **On the generative side**, we introduce a state-aware transformer with path-conditioning that ensures the generated motion conforms to the geometry and semantics of the scene. The model jointly conditions on both the path and the text description and outperforms baselines. **On the simulation side**, we propose phase-coherent ray tracing, which extends standard ray tracing to maintain consistent multipath geometry interaction and phase evolution across space and time for moving objects like humans and arbitrary radar configurations.
>
> In summary, WaveVerse introduces a new paradigm for RF signal simulation by unifying generative 4D worlds with physically grounded simulation. This paradigm, together with our technical innovations in scene-aware motion generation and coherent ray tracing, enables a level of diversity, realism, and generalizability that was not possible with prior approaches. We believe this direction opens new opportunities for scalable RF data generation, model training, and future research in RF sensing and simulation.

---

> ### Author Response · Authors · 2025-11-25
> **Author Response (2/3)**
>
> ### **Response to W.2: Data calibration**
>
> We thank the reviewer for raising this important point. We fully agree that real-data-driven calibration or refinement could further improve simulation fidelity, and we have discussed this as a promising direction for future work in Sec. 5 of our revised manuscript. At the same time, we have carefully ensured that the WaveVerse pipeline produces realistic and accurate signals. Beyond our proposed phase-coherent ray tracing, the simulator is tightly matched to the actual hardware: we follow the radar technical reference manual for antenna gain patterns and use the same radar configuration (including carrier frequency, slope, sampling rate, etc) as in real datasets.
>
> To further validate fidelity, we have included new quantitative evaluations measuring the similarity between simulated and real signals (the fourth paragraph of the High-Resolution Imaging (Sec. 4.4)). Specifically, we compute both *Fréchet Inception Distance* (*FID*) and the *Jensen–Shannon divergence of the TR margin* (*JS Div.*).  *FID* quantifies the distance between the feature distributions of real and simulated data, while *JS Div.* measures the gap between the confidence of a model on real data vs. synthetic data. For both metrics, a lower value indicates higher realism. We train a U-Net on the RF imaging task and compute *FID* on features extracted at the bottleneck, and compute *JS Div.* following [1]. Our results show that WaveVerse achieves an *FID* of 2.879 and a *JS Div.* of 0.365. These numbers are on par with those reported for strong generative models [2, 3, 1], and removing the modeling of phase coherence degrades performance substantially (*FID* 5.495 and *JS Div.* 0.430). Moreover, models trained on WaveVerse data transfer well to real measurements across tasks (Sec. 4.4), which also demonstrates the fidelity of the simulated signals.
>
>
> ---
>
> ### **Response to W.3: Inference overhead**
> We thank the reviewer for raising this question.  A detailed breakdown of the computation cost is provided in Appendix A.2, and we explicitly direct readers to this breakdown in the last sentence of Sec. 4.2 of the revised manuscript for clarity. For the reviewer’s convenience, we also summarize the numbers below. All numbers reported below are averaged over 10 runs on an RTX 3090 GPU and an i9-11900 CPU. WaveVerse is fully automatic and does not require human interaction, and its components can be parallelized across scenes and radar positions to improve throughput.
>
> **Input Prompt**. Generating scene and human-shape descriptions takes 1.37 s and 0.56 s, respectively, with most of the latency coming from OpenAI API communication rather than local computation.
>
> **Environment & Human Shape Generation**. Environment generation and human-shape generation take 105.47 s and 5.17 s, respectively. This includes API calls, mesh creation, object selection and placement in Unity, and loading the fine-tuned LLM checkpoint for human shapes. Importantly, this cost is incurred once per environment and can be amortized over many motion sequences and simulated signals.
>
> **Motion Description & Path Generation**. Generating motion descriptions and planning paths within the environment (again via API + path search) takes on average 7.03 s.
>
> **Human Motion Generation**. The Human Motion Generation module takes 20.79 s in total, although only 0.48 s comes from motion generation with our state-aware transformer. The dominant cost is SMPL fitting for the human mesh, which can be further optimized with faster implementations in computer vision.
>
> **Dielectric Property Generation**. Dielectric properties are precomputed, and the time is already included in environment generation.
>
> **Signal Simulation**. Our phase-coherent ray tracing simulates RF signals for a radar with 3 transmitters, 4 receivers, and 100k cast rays in 0.86 s.

---

> ### Author Response · Authors · 2025-11-25
> **Author Response (3/3)**
>
> ### **Response to W.4 & Q.1: Generalization and failure cases**
>
> We thank the reviewer for raising this important question. We agree that understanding when performance may degrade is crucial for assessing the real-world applicability of WaveVerse.
>
> We identify two scenarios where the sim-to-real gap can become more pronounced.
>
> 1).  **Fine-grained human–object interactions**.
>
> WaveVerse’s 4D generative pipeline models whole-body dynamics, which is sufficient for most RF sensing tasks, but it does not yet simulate detailed interactions such as typing or manipulating small objects. As a result, the applicability of WaveVerse in interaction-centric scenarios is still limited. However, as world-generation and motion-generation models continue to improve, WaveVerse, as a unified generation-and-simulation framework, can be naturally extended to handle fine-grained human-object interactions.
>
> 2). **Diffraction and Refraction**.
>
> Our simulation is built on ray tracing with reflection modeling, which dominates indoor RF propagation and supports most RF sensing tasks. However, more complex phenomena such as diffraction around sharp edges and refraction through objects are currently simplified, as in prior work [4,5]. Extending the simulator with UTD-based diffraction and Fresnel-based refraction is a promising direction to reduce this gap, and we leave this for future work.
>
> We have added a new discussion in Sec. 5 to reflect and highlight all these discussions in our paper.
>
> [1] Gong, Chen, et al. "Data Can Speak for Itself: Quality-guided Utilization of Wireless Synthetic Data." Proceedings of the 23rd Annual International Conference on Mobile Systems, Applications and Services. 2025.
>
> [2] Tian, Keyu, et al. "Visual autoregressive modeling: Scalable image generation via next-scale prediction." Advances in neural information processing systems 37 (2024): 84839-84865.
>
> [3] You, Zebin, et al. "Effective and efficient masked image generation models." arXiv preprint arXiv:2503.07197 (2025).
>
> [4] Cai, Hong, et al. "Teaching rf to sense without rf training measurements." Proceedings of the ACM on Interactive, Mobile, Wearable and Ubiquitous Technologies 4.4 (2020): 1-22.
>
> [5] Ren, Zhenyu, et al. "CASTER: A computer-vision-assisted wireless channel simulator for gesture recognition." IEEE Open Journal of the Communications Society 5 (2024): 3185-3195.

---

### Author Response · Authors · 2025-12-03
**Summary Remarks**

We thank the reviewers for their insightful feedback and valuable suggestions, which have helped us improve our paper during the rebuttal.

---

### **Recognized strengths**
Reviewers highlighted several positive aspects of WaveVerse:

1). **Novel RF Simulation Framework**: The design of a novel RF simulation framework with *4D world generation* and *a phase-coherent ray-tracer* was noted as an **innovative RF simulation framework** with **strong engineering effort** (hWWj, wPPx, Rnyj).

2).  **LLM Integration for Scalability**: The **innovative** use of LLMs for environment and motion generation was seen as an effective way to build **a scalable pipeline** that greatly improves data diversity. (hWWj, wPPx, A1nB, Rnyj).

3).  **Real-World Effectiveness**: The **comprehensive case studies** with **real-world sensing tasks** were viewed as **strong evidence** of the framework's effectiveness (wPPx, hWWj).

4).  **Bridging Communities**: The **conceptual innovation** of unifying *4D generative assets* with *RF rendering* was recognized as helping bridge the gap between computer vision and RF sensing and laying a foundation for future research (A1nB, Rnyj).

---

### **Key concerns and our responses**
We provided point-by-point responses to all concerns raised. Below, we summarize the main critiques and the actions we took to address them.

1). **Comparison with other simulation methods** (wPPx, Rnyj).

We add *RF Genesis* as a baseline for Human Activity Recognition and *standard ray tracing* for High-Resolution Imaging. Our method **consistently outperforms** the baselines across all levels of data augmentation, highlighting the advantages of our hybrid simulation approach.

2). **Accuracy and realism of the proposed RF simulation** (A1nB, Rnyj)

We substantially expanded our validation along three axes: (1) we added FID and JS Div. metrics to show close statistical alignment between simulated and real signals; (2) we replicated real-world scenarios with a synchronized camera–radar setup showing **28.63 dB PSNR** and **93.65% similarity**; (3) we compared WaveVerse against full-wave EM solver **HFSS**, achieving **33.57 dB PSNR / 2.12% RMSE** (w/o diffraction/refraction) and **31.25 dB PSNR / 2.76% RMSE** (w diffraction/refraction).
All results consistently show that WaveVerse produces accurate and realistic RF signals. The results are appreciated by reviewer A1nB.

3). **Plausibility of motion and material generation** (wPPx, A1nB, Rnyj)

We analyze the physical plausibility by evaluating **motion quality** via *Skating Ratio* and *Bone-Length Variance* metrics, showing close agreement with real motion statistics. We also clarify the **dielectric property generation**, where all material parameters are derived from ITU-R P.2040-2 standard; this process was noted by reviewer A1nB as appropriate and reassuring.

4). **Limitations and Future Work** (hWWj, A1nB)

We add a section discussing limitations (e.g., simplified diffraction/refraction modeling and fine-grained interactions) and outlining future extensions.

5). **Improved presentation and clarity** (hWWj, wPPx, A1nB)

We refine the manuscript for clarity, incorporating suggestions from the reviewer wPPx and adding references to the Appendix for details such as execution time and implementation choices.

---

### **Summary of online discussion**
- hWWj (rating 6, confidence 4): Our clarifications and additional analysis addressed all concerns raised by this reviewer, although further discussion was not possible.

- wPPx (rating 6, confidence 4): We appreciate the reviewer’s willingness to raise their score to 8 if the identified weaknesses were addressed. Although no further feedback could be provided, we believe the additional experiments and clarifications in this rebuttal successfully address all of the reviewer’s points.

- A1nB (rating 4, confidence 3): We interacted with the reviewer and were encouraged that most concerns were resolved. Following their final suggestion, we added additional results demonstrating the accuracy and realism of our simulation, which we believe fully address their remaining concerns.

- Rnyj (rating 4, confidence 4): We clarified the experimental setup, expanded technical details, and added new experiments in response to this reviewer’s concerns. Although no feedback was provided, we believe these revisions comprehensively address all issues raised.

---

### **Final note**

The strengths highlighted during the review process clearly reflect overall significance and potential impact of the work. The rebuttal period has further strengthened the paper across all major points raised by the reviewers. We are sincerely grateful for the reviewers’ thoughtful feedback and respectfully ask the AC to consider both the strengths recognized by the reviewers and the improvements made during the rebuttal. We hope WaveVerse will serve as a solid foundation for future work in RF sensing, simulation, and cross-modal generative modeling.

---

### Meta-Review · Area_Chair_L413 · 2025-12-27

**Summary:**

Reviewers generally find the paper conceptually novel and well engineered, proposing a scalable framework that combines generative 4D world modeling with physics-based RF simulation. The integration of language-guided scene and motion generation with phase-coherent ray tracing is viewed as innovative and promising for alleviating RF data scarcity.

However, reviewers consistently raised concerns about validation depth and technical novelty. Several questioned whether the contributions extend beyond a careful integration of existing components, and whether the physical realism of the simulated RF signals is sufficiently demonstrated, particularly without extensive scenario-level or paired real-world validation. Additional concerns include comparisons to existing RF simulators, physical plausibility of generated motions and materials, claim overstatement, and unclear computational overhead. While the rebuttal adds experiments and clarifications, some reviewers remain unconvinced about the fundamental limits of validating realism in fully synthetic 4D environments and the absence of formal error analysis.

**Reviewer Concerns:**

**Addressed Concerns**

- Lack of baseline comparisons (Raised by: wPPx, Rnyj):
The authors added comparisons against RF Genesis and standard ray tracing across multiple augmentation regimes, showing consistent gains.
- Physical plausibility of generated motion and materials (Raised by: wPPx, A1nB, Rnyj):
Quantitative motion realism metrics (FID, skating ratio, bone-length variance) and clarification of ITU-based dielectric modeling were added and generally accepted.
- Unclear claims and overstatement (e.g., “data-adequate”, “first RF imaging”) (Raised by: wPPx):
Claims were revised and terminology was softened.
- Inference/runtime overhead (Raised by: hWWj):
Detailed timing breakdowns were added to the appendix and referenced in the main text.
- Clarification of multipath modeling and simulator assumptions (Raised by: A1nB, Rnyj).

**Outstanding Concerns**

- Scenario-level physical validation against real measurements (Raised by: A1nB, Rnyj):
While controlled experiments and HFSS comparisons were added, reviewers note that such validation remains limited in scope and cannot fully resolve realism gaps for arbitrary generated worlds.
- Depth of core technical innovation (Raised by: hWWj, Rnyj):
Some reviewers remain unconvinced that the contributions go substantially beyond a well-engineered integration of existing generative and simulation components.
- Formal analysis of phase coherence and error bounds (Raised by: Rnyj):
Empirical demonstrations were added, but no formal analysis is provided.

**Reviewer Scores:**

- Reviewer hWWj (score: 6): Although the rebuttal addressed the raised concerns, this reviewer already viewed the work as marginally above threshold and emphasized limited core technical novelty.
- Reviewer wPPx (score: 6): While the reviewer indicated conditional openness to a higher score, the remaining uncertainties around physical validation and methodological depth suggest insufficient grounds for a definitive score increase.
- Reviewer A1nB (score: 4): Despite added metrics and controlled validations, this reviewer explicitly maintained that scenario-level physical validation remains a significant limitation.
- Reviewer Rnyj (score: 4): Clarifications improved presentation, but fundamental concerns about realism gaps and lack of formal analysis persist.

---

### Decision · Program_Chairs · 2026-01-26

Reject